# New and Old Horizons for an Ancient Drug: Pharmacokinetics, Pharmacodynamics, and Clinical Perspectives of Dimethyl Fumarate

**DOI:** 10.3390/pharmaceutics14122732

**Published:** 2022-12-06

**Authors:** Paolinelli Matteo, Diotallevi Federico, Martina Emanuela, Radi Giulia, Bianchelli Tommaso, Giacchetti Alfredo, Campanati Anna, Offidani Annamaria

**Affiliations:** 1Department of Clinical and Molecular Sciences-Dermatological Clinic, Polytechnic Marche University, 60121 Ancona, Italy; 2UO Dermatologia, INRCA/IRCCS, 60129 Ancona, Italy

**Keywords:** dimethyl fumarate, pharmacodynamics, pharmacokinetics, adverse effects, psoriasis, multiple sclerosis, immune-mediated diseases

## Abstract

(1) Background: In their 60-year history, dimethyl fumarate and other salts of fumaric acid have been used for the treatment of psoriasis and other immune-mediated diseases for their immune-modulating properties. Over the years, new mechanisms of action have been discovered for this evergreen drug that remains a first-line treatment for several different inflammatory diseases. Due to its pleiotropic effects, this molecule is still of great interest in varied conditions, not exclusively inflammatory diseases. (2) Methods: The PubMed database was searched using combinations of the following keywords: dimethyl fumarate, pharmacokinetics, pharmacodynamics, adverse effects, psoriasis, multiple sclerosis, and clinical indications. This article reviews and updates the pharmacokinetics, mechanisms of action, and clinical indications of dimethyl fumarate. (3) Conclusions: The pharmacology of dimethyl fumarate is complex, fascinating, and not fully known. Progressive insights into the molecule’s mechanisms of action will make it possible to maximize its clinical efficacy, reduce concerns about adverse effects, and find other possible areas of application.

## 1. Introduction

Characterized by high hygroscopicity and fungicidal power, dimethyl fumarate (dimethyl (2E)-but-2-enedioate—DMF) was once used as a dehumidifying agent and was placed in special bags, sometimes called ‘silica gel’, inside furniture or shoe boxes and leather bags to prevent deterioration during transport and storage in high humidity environments. DMF is the methyl ester of fumaric acid, whose name derives from the earth smoke plant *Fumaria officinalis*. The first description of a medical use dates to 1959, when the German biochemist Schweckendiek, who suffered from psoriasis, highlighted the benefits of fumaric acid esters (FAEs) on his inflammatory skin disease [1]. In 1994, a combination of FAEs containing DMF and salts of monoethyl fumarate was licensed for the treatment of psoriasis by the German Drug Administration with the name of Fumaderm^®^. The combination was empirically determined, as well as its dosing regimen. Since then, the action and contribution of the different FAEs to their overall therapeutic effect in psoriasis has been studied more in detail, showing that DMF is the main active agent via its metabolic transformation to monomethyl fumarate (MMF) [2]. Today, DMF is still a mainstay in the treatment of psoriasis, and over time its clinical indication has expanded to multiple sclerosis (MS). Moreover, during the current SARS-CoV-2 pandemic, DMF is experiencing renewed interest for the treatment of inflammatory immune-mediated diseases, not being a conventional immunosuppressant but rather a molecule with pleiotropic effects.

## 2. Pharmacokinetics

After oral ingestion, a great part of DMF is rapidly metabolized by esterase in the gastrointestinal tract into its primary, active metabolite MMF [3] (Figure 1). DMF is hardly detectable in systemic circulation and less than 0.1% of the dosage taken orally may be detected in urine [4]. As an α,β-unsaturated electrophilic compound, an of DMF is rapidly attacked by the detoxifying agent glutathione (GSH) by a Michael addition reaction prior to systemic distribution. Other fumarates, including MMF, can variably interact with free cysteine residues by Michael addition reaction, contributing to the modulation of availability of cellular GSH. Systemic concentrations of MMF peaks 2–2.5 h after the administration. Contemporary consumption of fatty food retards MMF systemic peak concentrations without modifying the Area Under the Curve [5]. This pharmacokinetic aspect has an important clinical implication, since co-administration with food increases the tolerability to potential adverse effects. The plasma protein binding of MMF is about 50%. In vitro studies have shown that the biotransformation of DMF does not involve cytochrome P450 enzymes and that MMF does not have any role as an inductor or inhibitor of this enzymatic complex [6]. Moreover, it does not interact with the P-glycoprotein system and with other common efflux and uptake transporters. For these properties, DMF has no major pharmacokinetic interactions. However, since diarrhea is a frequent side effect, clinicians should consider that concomitant absorption of other drugs may be reduced. MMF is mainly eliminated by exhalation of CO_2_, and only small amounts of the molecule are excreted through urine and feces. The apparent terminal elimination half-life of MMF is about 2 h.

## 3. Pharmacodynamics

The degree to which DMF and MMF can impact the involved pathways differentially has yet to be defined and is complicated by significant variation in the tissue exposure levels, absorption rate, and rate of metabolism for each of the various compounds. Based on the metabolism and pharmacokinetics of DMF, it is impossible to know the in vivo targets and local tissue exposure of DMF [3]. The pharmacodynamics of fumaric acid esters are not completely understood. DMF and MMF, as many other immunomodulators, seem to exert pleiotropic biological effects.

### 3.1. Enhancing Nuclear Factor 2 Pathway

One of the main pharmacological effects is related to the activation of the nuclear factor 2 (erythroid-derived factor 2, Nrf2) transcription pathway. This ubiquitous pathway works as a cell defense system against potential injuries derived from inflammatory and oxidative stress, both of which are considered triggers in the pathogenesis of immune-mediated inflammatory diseases such as MS and psoriasis. Nrf2-mediated cellular response is essential in the maintenance of the cellular redox homeostasis by upregulating the expression of antioxidant response element (ARE)-mediated genes. These include about 1% of human genes that encode a large variety of antioxidant enzymes and cytoprotective proteins [7]. Nrf2, under physiological conditions, is localized in the cytoplasm as a dimer of the Kelch-like ECH-associated protein (KEAP1). KEAP1 is a zinc finger protein that works as a redox sensor due to domains containing critical cysteine residues [8]. KEAP1 represents the main negative regulator of Nrf2 and, in a state of quiescence, induces the ubiquitination and proteolysis of Nrf2. In fact, KEAP1 serves as an adaptor for Cullin3 (Cul3)-based E3 ligase that regulates the proteosomal degradation of Nrf2. In the case of oxidation or alkylation of the highly reactive cysteine residues of KEAP1, as occurs in the presence of increased oxidative or electrophilic agents, the protein complex destabilizes and Nrf2 is no longer degraded [9]. Subsequently, Nrf2 moves into the nucleus as a heterodimer with small Maf proteins and acts as a transcription factor by binding the ARE sequence in the promoter of different target genes that encode for cytoprotective proteins, including heme oxygenase-1 (HO-1), quinoline oxidoreductase-1 (NQO1), catalase, superoxide dismutase (SOD), glutathione, and glutathione-dependent antioxidant enzymes [10]. In vitro studies have shown that DMF, as an electrophilic agent, oxidizes reactive thiols on KEAP1 and causes depletion of the intracellular antioxidant stocks of glutathione by forming conjugations with it, resulting in an enhanced activation of the Nrf2 pathway (Figure 2).

Therefore, the operating principle of DMF is related to the toxicological mechanism called “hormesis”, for which subtoxic amounts of potentially harmful oxidants or electrophiles can stimulate cell protection against toxic concentrations of the same or other molecules via the activation of the Nrf2 pathway. This phenomenon is analogous to the ischemic preconditioning, in which a sub-lethal ischemic insult has a protective effect on myocardial or brain cells against a subsequent ischemic challenge [11]. In vitro studies have shown that, even if to a lower degree, MMF also directly enhances Nrf2′s pathway but causes less intranuclear accumulation of the NFR2 protein compared to DMF [12]. Cuadrado et al. demonstrated that DMF could also stimulate Nrf2 activation by enhancing the PI3K/AKT/GSK-3β pathway [13]. Glycogen synthase kinase-3β is a serine-threonine kinase that phosphorylates critical residues of Nrf2, facilitating its ubiquitin-proteosome degradation. DMF modifies the phosphorylation status of GSK-3β and reduces its activity in the hippocampus of laboratory mice, resulting in a downregulation of this proteolytic pathway targeting Nrf2. Thus, DMF could exert a double mechanism of activation of Nfr2 by disrupting KAEP1/NRF2 and by reducing Nfr2 degradation [14]. DMF also appears to modulate TAU levels of phosphorylation in mouse models. TAU proteins stabilize microtubules, and when hyperphospholyrated cause severe neurodegenerative disease called taupathies, such as Alzheimer’s disease. GSK-3β regulates intracellular substrates involved in neuronal polarization, including collapsing response mediator protein 2 (CRMP2) and TAU, active players of microtubule organization. GSK-3β hyperactivity has been linked to this group of diseases, and DMF appears a promising approach in their treatment [15]. Finally, Nrf2 exerts a modulation of the nuclear factor kappa-light-chain-enhancer of activated B cells (NF-κB) pathway within a regulatory feedback loop, leading to a repression in the activity of this potent transcription factor of pro-inflammatory genes [16].

### 3.2. Effects on the Hydroxycarboxylic Acid Receptor 2 (HCAR2)

The cytoprotective effect of fumaric acid esters, especially on nervous tissue, can also occur due to the agonist action that DMF and, even more so, MMF exert on the HCAR2. This is a G protein-coupled receptor for nicotinic acid (niacin) expressed in many immune cells such as neutrophils, dendritic cells, macrophages, and microglia that, independent from Nrf2, activates a potent anti-inflammatory signaling. The HCAR2 downstream pathway leads to NF-κB inhibition, and subsequentially modulates the phenotype of microglia in the brain by promoting a switch towards an anti-inflammatory pattern of cytokine secretion and by normalizing the glutamatergic synaptic transmission [17]. This mechanism has shown a therapeutic effect on some experimental models of neurological diseases such as autoimmune encephalomyelitis, where DMF and MMF reduced neutrophil adhesion, migration, and recruitment to the central nervous system (CNS) [18]. The effects are not limited to the CNS but also involve other immune cells: HCAR2 activation on macrophages has been linked to an anti-atherogenic activity. HCAR2 also promotes the secretion of prostaglandins D2 and E2, whose effect of vasodilatation is responsible for mediating niacin and DMF-induced flushing, which is one of the most common adverse effects of the drug [19].

### 3.3. Modulation of GSH

Reactive oxygen species (ROS) have a critical role in the pathogenesis of neurological inflammatory disease and psoriasis. ROS induce transendothelial leukocyte migration with subsequent myelin and axonal damage. ROS production was found to be increased in the psoriatic plaque. GSH represents a main intracellular antioxidant system. The initial depletion of GSH caused by fumarates is not as severe to cause programmed cell death, but rather it promotes antioxidant mechanisms of compensation not only by boosting glutathione and glutathione-related enzymes de novo via the Nrf2 pathway, but also by enhancing GSH recycling, as demonstrated by Hoffman et al. [20]. GSH levels’ modulation caused by DMF could contribute to the inhibition of NF-κB in an Nrf2-independent manner [21].

### 3.4. Inhibition of NF-κB

The effect on NF-κB can be explained not only by the mechanisms described above, but indeed the inhibition of this important pro-inflammatory transcription factor results also from the interaction of DMF with the p65 protein, also known as the nuclear factor NF-κB p65 subunit. This molecule is an essential post-translational regulator of NF-κB, whose phosphorylation and acetylation are required for its activation. DMF, acting as an electrophilic agent, induces covalent modification of cysteine residues on p65, resulting in the blockade of the nuclear translocation and phosphorylation of this protein. This DMF-mediated mechanism of inhibition of NF-κB has been demonstrated by Kastrati et al. in breast cancer cells and, added to the activation of the Nfr2 cytoprotective pathway, may suggest an anti-tumor activity of DMF [22].

### 3.5. Effects on the Immune System

In vitro studies by Treumer et al. demonstrated a proapoptotic action of DMF on human purified T-lymphocytes in a concentration- and time-dependent manner [23]. Apoptosis is the consequence of GSH depletion induced by DMF that, in some target cells, may reach the threshold for induction of cell death. The mechanism was found to be more intense on activated T cells when compared to resting T cells. Indeed, during DMF therapy lymphopenia represents one of the major adverse effects, and complete blood count needs to be monitored periodically. However, this side effect only affects a minority of patients, thus it is possible that only in some predisposed individuals DMF reaches high tissue concentrations, which perhaps combined with other intercurrent lymphocytotoxic stimuli induces massive T cell death. In addition to the possible effect on leukocyte count, fumarates modulate the inflammatory profile of T cells by inducing a shift from the Th1 to Th2 lymphocyte phenotype and cytokine pattern. Th1-activated lymphocytes with their secretion of interferon γ (IFN-γ) and interleukin (IL)-17 play an important role in the pathogenesis of both psoriasis and MS. In vitro studies by de Jong et al. demonstrated that MMF increased the production of cytokines related to Th2 lineage such as IL-4 and IL-5 in stimulated T cells, without suppressing the production of Th1 cytokines [24]. Another investigation has shown that DMF reduces the differentiation of T naïve cells’ into reactive Th1 and Th17 lymphocytes by inhibiting dendritic cell maturation. The inhibition of the pro-inflammatory cytokines downstream of the NF-κB pathway has an important role in rebalancing these two branches of the adaptative response. As a result of this anti-inflammatory activity, DMF caused inhibition of the tumor necrosis factor (TNF)-α-induced expression of the adhesion molecule, as did E-selectin and VCAM-1 in an experimental model of endothelial cell cultures, leading to the reduction in leukocyte recruitment [25]. The psoriatic plaque is characterized by a skin accumulation of lymphocytes due to upregulation of endothelial cell adhesion molecules, and this could explain one possible mechanism of action of DMF in this inflammatory disease [26].

### 3.6. Regulation of Iron Metabolism in the Brain

Iron is an essential element for myelination [27], and disorders of iron metabolism have been recently incriminated as possible triggers of neurodegenerative diseases. Extracellular unbound iron accumulation in the brain is an age-related phenomenon, but elevated iron levels in microglia can be considered a hallmark of pathological processes [28]. In physiological conditions, microglia cells store iron-bound ferritin and release it to oligodendrocytes to support myelination [29]. In MS, there is a pathological unbound iron release by damaged oligodendrocytes in the extracellular space that induces the production of free radicals and accumulates in macrophages and microglia. The subsequent activation of the inflammatory cascade results in neuron damage and disruption of the myelination process [30]. Iron accumulation in macrophages and microglia around the rim of the demyelinated plaques in MS can be highlighted by magnetic resonance imaging (MRI) [31]. Evidence in primary murine microglia shows that DMF acts as a regulator of the brain’s iron metabolism by upregulating specific proteins (such as T-cell immunoglobulin mucin domain 2 protein (TIM-2) and transferrin receptor 1) that improve the uptake of iron in microglial cells and the transfer of ferritin to oligodendrocytes and promote a shift of the microglia towards an anti-inflammatory and neuroprotective phenotype [32].

### 3.7. Antimicrobial Effects and Modulation of Gut Microbiota

Fumaric acid esters, as organic acids, are commonly used as antimicrobials in foods [33]. Although it has never been used in humans for its antibiotic activity, DMF exerts an antimicrobial activity towards different microorganisms. An antibacterial effect against Escherichia coli has been demonstrated in vitro by Wang et al. [34] and against Clostridium perfringens by Rumah et al. [35]. DMF in vivo also seems to modulate the gut microbiome profile. This is possible, maybe not only due to its antimicrobial effect, but also because some intestinal bacteria are able to metabolize fumaric acid esters with a competitive advantage [36]. A treatment of 12 weeks with oral DMF in patients with MS was not associated with significant alterations in the gut microbiota composition but showed a near-normalization of the low abundance of butyrate-producing Faecalibacterium seen in MS patients [37]. Butyrate is an important nutrient for the homeostasis of the gut barrier and the intestinal immune system, exerting a beneficial anti-inflammatory effect in intestinal and extraintestinal disease. Thus, a direct effect on the gut microbiota could account for the therapeutic action of DMF.

## 4. Adverse Effects

### 4.1. Gastrointestinal Adverse Effects

Given the pharmacokinetics of DMF, it is known that only a small fraction of the molecule enters the systemic circulation after oral administration, and it is possible that this small amount is responsible for the hermetic dose response in the Nrf2 pathway that is observed in neurons or other tissues exposed to a low concentration of the drug. The gut barrier, on the other hand, is exposed to a high concentration of the drug, which would cause large oxidative stress and GSH depletion. This could explain the high frequency of gastrointestinal (GI) adverse effects in patients treated with DMF and the warning not to consume alcohol with the drug to improve tolerability. In fact, combining several harmful stimuli at the same time would exponentially amplify the toxic effect on the GI epithelium, while taking the drug with food (particularly a high-fat meal) reduces the incidence and severity of GI adverse effects. Diarrhea is the most common adverse reaction, followed by abdominal distension and pain, nausea, vomiting, dyspepsia, and constipation. These side effects usually occur in the first 2 to 3 months of treatment, after which a gradual tolerance develops. They are usually mild to moderate, although diarrhea could become so severe to affect the absorption of other medicinal and oral contraceptives. There are no apparent dose-related effects and no risk factors for the occurrence could be identified, however, slow dose titration when initiating therapy is useful to reduce GI discomfort. Furthermore, specific symptom-directed therapies are useful to manage intolerable side effects (Table 1) [38]. Although there are experimental studies that show anti-inflammatory responses in immune-mediated colitis, severe gastrointestinal disorders remain a contraindication in the use of the drug.

### 4.2. Hematological Disorders

DMF may decrease leukocyte and lymphocyte counts. Notably, the risk of DMF-associated lymphopenia increases with age. In adults older than 55 years this side effect is more frequent and more severe, with grade 2 (500–799 cells/μL) and 3 (200–500 cells/μL) lymphopenia developing in more than 40% of cases in this age group. No cases of lymphopenia were observed in pediatric populations [39]. Patients with MS switched to DMF from natalizumab and those with a lower baseline lymphocyte count have a higher risk of developing this side effect. Thus, prior to initiating DMF therapy it is mandatory to perform a complete blood count (CBC), and the treatment should not be initiated if leukopenia below 3.0 × 10^9^/L or lymphopenia below 1.0 × 10^9^/L are reported. Blood monitoring should be performed after one month from the start and then every 3 months. If the lymphocyte count falls below 1.0 × 10^9^/L but is ≥ 0.7 × 10^9^/L, blood monitoring should be performed monthly until levels return to 1.0 × 10^9^/L or higher for two consecutive blood tests. If leukocyte count decreases below 3.0 × 10^9^/L or lymphocytes below 0.7 × 10^9^/L the treatment must be stopped immediately, and scrupulous monitoring of the count should be performed until it has returned to the normal range. Recovery after drug withdrawal is usually rapid (2–4 months) but it may be prolonged (≥6 months), especially in patients who reached lower lymphocyte count at DMF interruption. Moreover, this collateral effect was found to be more persistent in MS patients with longer disease duration and previous exposure to other immunomodulating treatments [40]. Patients who present severe and prolonged lymphopenia are at increased risk of developing progressive multiple leukoencephalopaty (PML) [41], even if it has also been reported in patients without severe lymphocytopenia [42].

### 4.3. Progressive Multiple Leukoencephalopathy

Progressive multiple leukoencephalopathy (PML) is a demyelinating disease caused by the John Cunningham virus (JCV), an opportunistic pathogen. JCV is a type of polyomavirus that infects approximately 70 to 90% of the general population [43] asymptomatically. In the case of immune deficiency such as HIV infections, chemotherapy, patients with transplants, and therapy with immunosuppressants, the virus can cause potentially fatal nervous system injuries. The most frequent symptoms are weakness, cognitive impairment, personality changes, and visual and speech disorders. The incidence of PML in patients with MS during treatment with DMF is 0.02 per 1000 patients, and the risk is higher in patients with persistent severe lymphopenia and older age, although cases have occurred in young patients with normal absolute lymphocyte count (ALC), suggesting that changes in specific subsets might be more important than total ALC [44]. In patients with psoriasis there are no specific data, although in a large, population-based study including subjects with psoriatic arthritis, psoriasis, rheumatoid arthritis, and other rhematic diseases, the incidence of PML was 0.2 per 100,000 persons without HIV or cancer. Most of the cases occurred in patients previously treated with rituximab and infliximab, although a causal role for biologic agents remains uncertain [45]. The diagnosis of PML requires a high index of clinical suspicion, especially in patients with a pre-existent neurological disease such as MS who could present natural fluctuations in the activity of the disease. A polymerase chain reaction (PCR) on cerebrospinal fluid (CSF) and an MRI are essential to corroborate the suspect and interrupt potentially harmful therapies [46].

### 4.4. Fanconi Syndrome

Fumaric ester acids can cause drug-induced Fanconi syndrome (FS). This is a disfunction of proximal renal tubules that can lead to inadequate reabsorption of glucose, bicarbonate, amino acids, and phosphate. Patients who develop FS typically present generalized weakness, myalgia, proteinuria and glycosuria, osteomalacia, and pathological bone fractures. The mechanism by which DMF appears to cause tubular damage, although unclear, appears to be related to mitochondrial damage, as shown by electron microscopy studies [47]. Although FS is an infrequent side effect, renal function (e.g., creatinine, blood urea nitrogen, and urinalysis) should be checked prior to initiation of treatment and every 3 months thereafter.

### 4.5. Flushing

Differently from FS, flushing is a very common adverse effect of DMF. HCAR2 is thought to be involved in the flushing reaction. When MMF binds to HCAR2, it induces prostanoid-forming enzymes in different cell types. Prostaglandin-induced vasodilatation may be mitigated by pretreatment with acetylsalicylic acid and slow titration of the dose [48].

## 5. Indications for DMF

### 5.1. Psoriasis

DMF is approved for moderate to severe chronic plaque psoriasis (CPP) as the first-line systemic therapy before starting a biological therapy. The first fumarate Fumaderm^®^ (a mixture comprising 60% DMF and calcium, zinc, and magnesium salts of monoethyl fumarate) has been available in Germany since 1994, and more recently a new formulation containing only DMF (Skilarence^®^) was approved for psoriasis in 2008 by the European Medicines Agency. The therapeutic dose of the drug ranges between 30 and 720 mg. An Italian real-life multicentric retrospective (30-month period) study published in 2021 including 103 psoriatic patients showed interesting benefit/risk ratio results for a first-line therapy: in particular, 23% of patients reached PAS75 at week 12, while more than 40%, or almost 80% considering the patients who were continuing the treatment, reached it at week 26 [49]. Side effects, mainly mild to moderate transitory gastrointestinal symptoms, were the main cause of treatment discontinuation. Slow titration of the dose was needed for a good tolerability of the drug, and when side effects appeared after a temporary dose reduction a subsequent further increase was often better tolerated by many patients. Most of the dropouts occurred in the first 5 months of treatment. With reference to variables associated with DMF effectiveness, patients who achieved higher values of PASI at T12 had a lower BMI and were taking a higher daily dose than those who did not. At the same time, treatment discontinuation was inversely related to the median daily dose and the BMI. Moreover, the study highlights the safety of the molecule in oncological patients, as a consistent number of patients treated had an oncological history (18.4%) and none of them had recurrence of malignancy during the follow-up. Data from either prospective interventional (BRIDGE) or non-interventional (DIMESKIN 1, SKILL) studies among patients with moderate-to-severe psoriasis showed that DMF also provides a positive efficacy profile in all four body regions included in the Psoriasis Area and Severity Index assessment (head and neck, trunk, and upper and lower extremities) and a particularly interesting profile (strong efficacy) in the head and neck region. These findings may be of special interest to patients with scalp psoriasis, which has always been considered one of the most difficult-to-treat areas. Patient-reported outcomes (quality of life and pruritus) also improved during the 24 weeks of DMF treatment [50].

Furthermore, DMF, due to its mechanism of action, could be considered an interesting therapeutic choice for patients suffering from psoriasis with chronic infections (e.g., HIV and TBC), whose frailty could be addressed by restoring a balance between the Th1/Th2 pathways [51,52]. In times of economic constraints, a study from the UK, performing a cost-effectiveness analysis of treatment sequences, confirmed the role of DMF prior to biological agents as a dominant treatment strategy for the treatment of moderate-to-severe psoriasis [53]. In conclusion, DMF appears as an effective, easy handling, and dynamic first-line therapy for CPP with a high cost-effectiveness ratio.

### 5.2. Multiple Sclerosis

The first observations that fumaric acid esters might have a role in the treatment of MS date back to 2006, when Fumaderm^®^ was administered to 10 patients in an open-label pilot study with encouraging results on both clinical and radiological progression of the disease [54]. Since then, there have been many clinical trials with fumarates on this autoimmune neurological disease. Currently, BG-12 (marketed as Tecfidera^®^), an orally administered, enteric-coated micro tablet slow-release preparation of DMF, is indicated for the treatment of this disease in the relapsing remitting variant (RRMS). RRMS is the most common subtype, characterized by unpredictable flare-ups followed by periods of months to years of incomplete or complete recovery, termed remissions. This formulation of DMF demonstrated fewer GI-related side effects compared to the old fumarates and is effective in reducing the number of new demyelinating lesions and annual relapses compared to the placebo [55]. Larger phase III group studies later confirmed the significant efficacy and the favorable benefit–risk profile of DMF in patients with RRMS failure to interferon beta. Over their 2-year duration, these multicenter, placebo-controlled, double-blind clinical trials proved that BG-12 reduced the number and the size of neurological lesions (observed via MRI) and significantly decreased the proportion of relapsing patients [56]. Although adverse effects such as nausea, vomiting, dyspepsia, diarrhea, and flushing were commonly reported, especially in the first weeks, a progressive improvement was observed during the treatment, and safety and tolerability of DMF were deemed acceptable in these trials [57].

## 6. Special Issues and Possible Future Indications

### 6.1. Pregnancy and Breastfeeding

DMF is not recommended in pregnant and breastfeeding women and should be used only if the benefit justifies the potential fetal risk. However, an interim analysis of 345 patients exposed to DMF during the first trimester in a prospective international registry performed by Hellwig et al. demonstrated that pregnancy outcomes (including gestational size, pregnancy loss, birth defects, and infant or maternal death) were not affected by the drug (with a mean exposure of 4.9 weeks) [58]. Moreover, due to its extremely short half-life, DMF can be considered a versatile drug to use in potential childbearing women, because the prompt interruption in the case of unexpected pregnancy could prevent possible fetal consequences. No information is available on the clinical use of dimethyl fumarate during breastfeeding. However, amounts of the active metabolite of dimethyl fumarate, monomethyl fumarate, appear to be low in breastmilk and would not be expected to cause any adverse effects in breastfed infants. Before any data were available, some authors recommend avoiding breastfeeding during dimethyl fumarate therapy [59].

### 6.2. SARS-CoV-2 Infection

Since the beginning of pandemic of SARS-CoV-2 in 2019, clinicians of all specialties must deal with the issue of treating inflammatory diseases with immunosuppressive drugs that could potentially predispose the patient to an increased risk of acquiring the infection or a more severe course. This is especially important when the patient is elderly, frail, or has multiple comorbidities [60]. The possibility to rely on a pleiotropic drug without pure immunosuppressive features such DMF is important in the clinician armamentarium. Moreover, DMF has demonstrated to exert a potent antiviral and anti-inflammatory activity through the activation of the NRF2-pathway, distinct from the type I interferon pathway, towards Herpes simplex 1 and 2, Vaccinia virus, Zika virus, and SARS-CoV-2 [61]. Experimental data from a small case series showed that MS patients treated with DMF showed developed self-limiting benign SARS-CoV-2 infection and could continue the therapy with DMF at the usual dosage [62]. An Italian multicenter retrospective study including 44 psoriatic patients who were taking DMF during 48 weeks of the pandemic showed that DMF was effective as monotherapy, and the patients who became positive to SARS-CoV-2 developed a mild form that did not require discontinuing the drug [63]. This protective effect could probably be related to the antioxidative and anti-inflammatory properties of DMF, which could counteract the cytokine storm caused by severe SARS-CoV-2 infections [64]. These data suggest that continuing DMF might be safe in patients with mild forms of SARS-CoV-2 infection and a normal lymphocyte count [65].

### 6.3. Cancer

Clinical and pre-clinical investigations have demonstrated that DMF exerts anti-tumorigenic properties in several types of cancer [66]. DMF was found to inhibit the proliferation of melanoma cells and tumor progression in mouse models, due to the inhibition of pro-metastatic matrix metalloproteinases and its proapoptotic action [67]. The suppression of the NF-κB pathway, which is abnormally activated in neoplastic cells and promotes their survival, migration, invasion, and resistance to chemotherapy, appears to be a crucial property of DMF in counteracting various types of cancer, including breast cancer and cutaneous T-cell lymphoma (CTCL) [22]. Notably, DMF has proved promising results and a good tolerability in CTCL patients in a phase II clinical trial (NCT02546440) [68]. DMF has also demonstrated cytotoxic effects towards other cancer cell lines, such as colorectal cancer [69], cervical cancer [70], and adenocarcinoma of the lung and pancreas [71]. This evidence suggests that DMF can be safely utilized in patients with a history of oncological disease.

### 6.4. Cardiovascular Diseases

The antioxidant and anti-inflammatory effects of DMF might play a role in the treatment and prevention of multiple cardiovascular diseases. DMF has conferred anti-atherosclerotic effects in patients with MS and psoriasis by improving the lipidic serum profile [72,73] and by elevating the levels of adiponectin [74]. Along with the hypolipidemic action, DMF seems to reduce the lipidic peroxidation in the arteries [75]. Whilst other studies suggest that FAE might be beneficial for patients with stroke, myocardial ischemia, hypertension, and aneurysms, more research is warranted to establish DMF’s efficacy for these indications [76].

### 6.5. Solid Lipid Nanoparticles for DMF Administration

DMF has been demonstrated to be useful in RRMS treatment. Nevertheless, since DMF administration can be associated with flushing, GI events, and other more serious drawbacks, Esposito E. et al. [77] investigated an alternative method of DMF administration, a nanoparticle-based system, to minimize side effects. The authors described the preparation and characterization of DMF-containing solid lipid nanoparticles (SLN). Namely, SLN based on tristearin, tristearin SLN treated with polysorbate 80, and cationic SLN constituting tristearin in a mixture with dimethyldioctadecylammonium chloride were investigated. The effects of the presence of DMF and functionalization by polysorbate 80 and dimethyldioctadecylammonium chloride were studied on the morphology and dimensional distribution of SLN by photon correlation spectroscopy and cryogenic transmission electron microscopy. DMF released from SLN, studied by Franz cell, evidenced a Fickian dissolutive-type kinetic in the case of SLN treated with polysorbate 80. The presented data indicate that DMF can be conveniently and efficiently encapsulated in SLN with dimensional and morphological properties well suitable for clinical applications requiring different administration routes. Microencapsulation of DMF in SLN could allow a significant improvement in drug absorption and the avoidance of first-pass effects. The results reported by the authors represent a promising starting point for the further development of an SLN-based DMF formulation for the treatment of MS.

## 7. Conclusions and Expert Opinion

DMF is a drug with a long history, which has proved to have pleiotropic pharmacological effects that could make it useful in multiple conditions including inflammatory, degenerative, neoplastic, and cardiovascular diseases. Since it is not a classical immunosuppressant, it is of particular interest in the current time of the SARS-CoV-2 pandemic. The lack of major pharmacokinetic interactions makes it an interesting drug in elderly and multi-comorbid patients. Moreover, the short half-life and recent observations on pregnancy suggest that it could be an interesting, safe option in fertile women. It has probably been underexploited in dermatology, especially in some countries such as Italy, due to concerns about its potentially serious adverse effects and a lack of familiarity with the dosage of the drug. However, good knowledge of its pharmacological features, slow titration of the dosage, and cautious monitoring of patients during therapy can maximize its therapeutic window. In a clinical setting dominated by increasingly expensive drugs, DMF could represent an inexpensive and effective option. Further understanding of its pharmacodynamics and large randomized trials in the future could better define its clinical indications and safety profile.

## Figures and Tables

**Figure 1 pharmaceutics-14-02732-f001:**
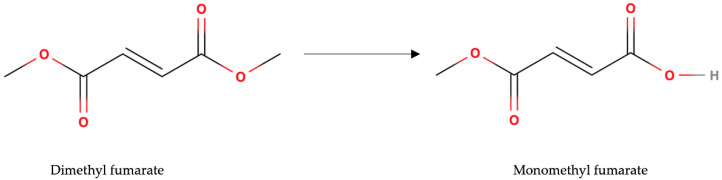
Pre-systemic metabolism of dimethyl fumarate to monomethyl fumarate by esterase.

**Figure 2 pharmaceutics-14-02732-f002:**
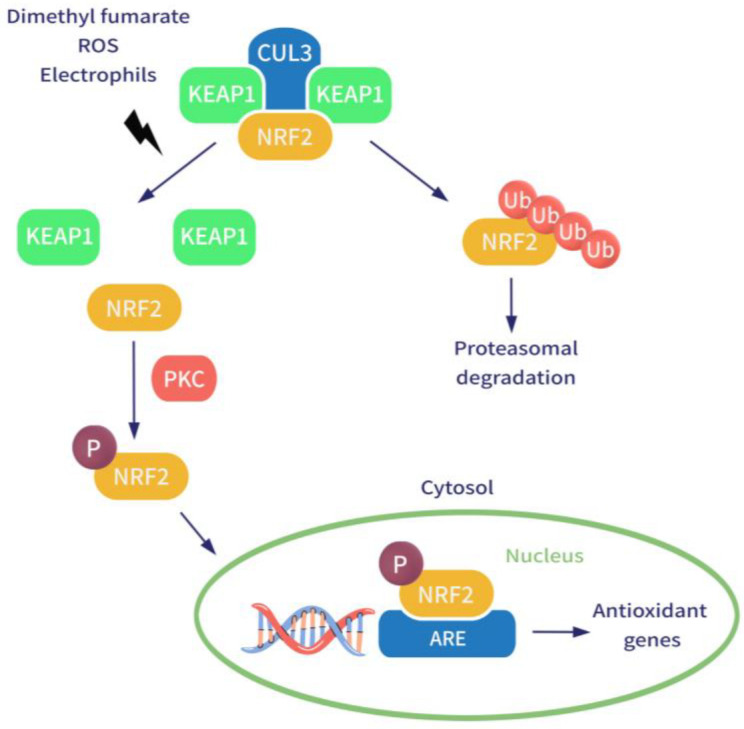
Effect of dimethyl fumarate on Nrf2 pathway. Abbreviations: NRF2, nuclear factor (erythroid-derived 2)-related factor; ARE, antioxidant response element; CUL3, Cullin3; KEAP1, Kelch-like ECH-associated protein; PKC, protein kinase C; Ub, ubiquitin; ROS, reactive oxygen species.

**Table 1 pharmaceutics-14-02732-t001:** Table with a list of the most frequent gastrointestinal adverse effects caused by DMF and possible therapeutic options.

Adverse Effects	Possible Therapeutic Options
Diarrhea	Loperamide
Abdominal pain	Proton pump inhibitors Bismuth subsalicylates H_2_ blockers
Vomiting	Ondansetron Promethazine
Nausea	Ondansetron Antiacids Promethazine Bismuth subsalicylates

## Data Availability

Not applicable.

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
