# Peer review of "New and Old Horizons for an Ancient Drug: Pharmacokinetics, Pharmacodynamics, and Clinical Perspectives of Dimethyl Fumarate"

_pharmaceutics, 2022, doi:10.3390/pharmaceutics14122732_

Round 1

Reviewer 1 Report

The present review manuscript entitled "New and old horizons for an ancient drug: pharmacokinetics, pharmacodynamics and clinical perspectives of dimethyl fumarate” involved the bibliographical analyses of DMF (historical drug) and its mechanisms of action, pharmacokinetics, and clinical indications.

The introduction is according to the aim of this research, and it is appropriate for the desire analyses/conclusions the authors reported but some information regarding the impact of this drug in the SARS-CoV-2 is not reported (in the introductory paragraph).

I suggest including clear and logical images/figures/tables to summarize the information. These improvements will allow the reader to understand the information easily and quickly throughout the manuscript. As an example: one figure to describe the Nrf2 pathway or some summarizing figure with all the mechanism of action.

Additionally, it will be interesting to include the specific IUPAC name for this compound due to its unique stereochemistry (trans or E isomer).

Besides, I encourage the authors to check some mistakes such as (highlighted in yellow, pdf attached):

- Please remember to use italics for in vivo, in vitro.

- Some typo mistakes should be corrected (lines 262, 264, 265, 266) when the scientific notation is used and lines 326 should be period (30-month period).

- Furthermore, the international abbreviation for the severe acute respiratory syndrome coronavirus 2 is SARS-CoV-2, please correct them in the item 6.2 and conclusions.

Also, it should be interesting to analyze if there are some nanosystem which include this drug as drug delivery system.

Finally, I would like to invite the authors to include the abbreviation list of words at the end of this manuscript.

Author Response

Dear Reviewer,

thank you for your suggestions.

Point 1: The introduction is according to the aim of this research, and it is appropriate for the desire analyses/conclusions the authors reported but some information regarding the impact of this drug in the SARS-CoV-2 is not reported (in the introductory paragraph).

response 1: We add in the introduction (at line 44) a comment about the role of DMF in SARS-Cov-2 pandemic.

Point 2: I suggest including clear and logical images/figures/tables to summarize the information. These improvements will allow the reader to understand the information easily and quickly throughout the manuscript. As an example: one figure to describe the Nrf2 pathway or some summarizing figure with all the mechanism of action.

response 2: We include an image summarizing the effect of DMF in NRF2 pathway (figure 2).

Point 3: additionally, it will be interesting to include the specific IUPAC name for this compound due to its unique stereochemistry (trans or E isomer).

response 3:  With regard to IUPAC nomenclature, we add at line 29 the preferred name for DMF which is Dimethyl (2E)-but-2-enedioate; other name:  trans-1,2-Ethylenedicarboxylic acid dimethyl ester (E)-2-Butenedioic acid dimethyl ester.

Point 4: Besides, I encourage the authors to check some mistakes such as (highlighted in yellow, pdf attached):

- Please remember to use italics for in vivo, in vitro.

- Some typo mistakes should be corrected (lines 262, 264, 265, 266) when the scientific notation is used and lines 326 should be period (30-month period).

- Furthermore, the international abbreviation for the severe acute respiratory syndrome coronavirus 2 is SARS-CoV-2, please correct them in the item 6.2 and conclusions.

response 4: We correct the mistakes highlighted.

Point 5: Also, it should be interesting to analyze if there are some nanosystem which include this drug as drug delivery system.

Response 5: we add a paragraph (6.5) focusing encapsulated DMF into nanoparticles for the treatment of MS. This is a promising starting point for the further development of an SLN-based DMF formulation for the treatment of MS.

Point 6: Finally, I would like to invite the authors to include the abbreviation list of words at the end of this manuscript.

Response 6: we add a list of the principal abbreviations at the end of the document.

Kind regards

Reviewer 2 Report

The review article by Matteo et al. gives a nice overview on DMF and its potential and existing applications. In general, I would like to recommend publication once some issues have been adressed:

1) the authors mention the michael addition reaction with GSH. What about the possibility of a micheal addition reaction to the other mentioned targets? Also MMF still is a michael acceptor. This should be discussed.

2) I was wondering about the usefullness of table 1. I think treating adverse drug effects with other drugs only makes sense in some cases. Taking atropine to treat DMF-associated diarrhea would cause even more severe adverse effects.

3) The authors might think to include some graphical material or an additional table for the pharmacodynamics paragraph.

Author Response

Dear Reviewer,

thank you for your suggestions. 

Point 1: the authors mention the michael addition reaction with GSH. What about the possibility of a micheal addition reaction to the other mentioned targets? Also MMF still is a michael acceptor. This should be discussed.

Response 1: at line 55 we add a comment about this issue. 

Point 2: I was wondering about the usefullness of table 1. I think treating adverse drug effects with other drugs only makes sense in some cases. Taking atropine to treat DMF-associated diarrhea would cause even more severe adverse effects. 

Response 2: we delete atropine as a possible medication for DMF-induced diarrhea.

Point 3: The authors might think to include some graphical material or an additional table for the pharmacodynamics paragraph.

Response 3: we add an image summarizing the effect of DMF on NRF2 pathway (figure 2).

Kind regards
